# Comparative Analysis of Anti-Müllerian Hormone Concentration in Two Indigenous Slovenian Sheep Breeds

**DOI:** 10.3390/ani15091332

**Published:** 2025-05-05

**Authors:** Nataša Šterbenc, Janko Mrkun, Špela Petročnik, Meta Sterniša, Maja Zakošek Pipan

**Affiliations:** 1Clinic for Reproduction and Large Animals, Veterinary Faculty, University of Ljubljana, 1000 Ljubljana, Slovenia; natasa.sterbenc@vf.uni-lj.si (N.Š.); janko.mrkun@vf.uni-lj.si (J.M.); spela.petrocnik@vf.uni-lj.si (Š.P.); 2Department of Food Science, Biotechnical Faculty, University of Ljubljana, 1000 Ljubljana, Slovenia; meta.sternisa@bf.uni-lj.si

**Keywords:** anti-Müllerian hormone, sheep, reproductive activity, Jezersko–Solčava breed, Istrska pramenka breed, aging, ovarian reserve, prolificacy, fertility

## Abstract

Anti-Müllerian hormone is expressed in females only in granulosa cells, mainly by growing follicles. It serves as a biomarker for ovarian reserve in various mammalian species, including sheep. This study compares anti-Müllerian hormone levels in two indigenous Slovenian sheep breeds: Jezersko–Solčava and Istrska pramenka. A total of 78 sheep were included in the study and were grouped by age. Anti-Müllerian hormone levels in the blood varied significantly between the breeds, with the Jezersko–Solčava sheep showing higher concentrations. No age-related differences were observed within the Jezersko–Solčava breed. However, Jezersko–Solčava sheep over 7 years of age had higher anti-Müllerian hormone levels compared with Istrska pramenka sheep of the same age. Additionally, anti-Müllerian hormone levels were higher in ewes that lambed twins. This research underscores how breed and age affect anti-Müllerian hormone levels, as well as their possible significance in predicting reproductive success.

## 1. Introduction

Anti-Müllerian hormone (AMH) has become an important biomarker in female reproductive biology and serves as an indirect measure of ovarian reserve in various mammalian species, including sheep [1]. Ovarian reserve, originally conceptualized to describe the pool of primary follicles in the ovaries, has evolved into the total number of healthy ovarian follicles and oocytes. The dynamic nature of the ovarian reserve is of central importance, particularly given its finite nature in female mammals, where the progressive depletion of oocytes with age eventually culminates in the cessation of the reproductive capacity [2].

In recent years, AMH has gained importance as a reliable indicator of ovarian reserve due to its positive association with the number of antral follicles [3,4,5]. AMH production is primarily attributed to the granulosa cells of healthy, growing ovarian follicles in females, cattle, and sheep, and thus reflects the reservoir of follicles available for development [6,7,8,9,10]. Extensive studies in cattle have established AMH as a robust marker of ovarian reserve, with peripheral AMH concentrations reflecting the antral follicle population [11,12,13]. Similarly, studies in sheep and goats have confirmed the utility of AMH as an indicator of ovarian reserve, further emphasizing its consistent role across species [9].

A characteristic feature of AMH is its selective production, which begins with follicle recruitment and peaks during the preantral and small antral follicle stages. This production pattern is followed by a decline when the selected follicle enters the preovulatory stage under the influence of follicle stimulating hormone (FSH) [14,15]. This temporal regulation emphasizes that AMH is a marker specifically associated with the actively growing and healthy follicular cohort within the ovarian reserve [1].

Serum AMH concentrations exhibit considerable variability between individuals, but show a high degree of repeatability within the same animal, emphasizing the reliability of AMH as a consistent indicator of ovarian reserve status [16]. This property makes AMH particularly valuable for the assessment of the reproductive potential and the prediction of fertility outcomes in different mammalian species.

There are four indigenous sheep breeds in Slovenia, of which Istrska pramenka (IP) and Jezersko–Solčava (JS) are the most widespread. The IP breed, which is characterized by its seasonal reproductive activity and the JS breed, which is known for its year-round reproductive activity, provides an interesting comparative context for understanding the interplay between two different reproductive patterns. By quantifying the AMH levels in animals of both breeds, the study aims to shed light on breed-specific variations in AMH levels and their relationship with reproductive performance. The aim of this study was to determine AMH concentration in IP and JS sheep breeds, and to investigate the possible relationship between AMH concentration, age, breed, and fertility outcomes (number of lambs).

We hypothesized that the AMH concentration would decline with age and was correlated with fertility outcomes; however, the AMH levels would not significantly differ between the breeds.

## 2. Materials and Methods

### 2.1. Animals and Sampling

The animals of both breeds were kept at pasture from early spring until the winter months, with unrestricted access to water. During the winter, they were housed indoors and provided a hay-based diet. All of the selected ewes were clinically healthy, and routine deworming was performed on all animals. A breeding ram remained continuously with the herd to facilitate natural mating. Blood samples were collected uniformly between 08:00 and 10:00 a.m., during the confirmed estrus phase. Sampling for the JS sheep was conducted in the fall of 2022, while the IP sheep samples were obtained during the fall 2021 breeding season. Estrus was detected using a ram equipped with a marking harness. A total of 78 sheep from two different breeds were sampled for this study. From the IP breed (bred on the Vremščica plateau), 35 sheep were sampled, and from the JS breed (bred in Jezersko), 43 sheep were sampled. Due to the considerable variation in the age structure of the JS sheep, they were categorized into three age groups; A = 1–3 years (10 sheep), B = 4–6 years (18 sheep), and C = 7 years or older (15 sheep). A blood sample was collected once from each animal via jugular venipuncture to determine the AMH concentration. Fertility results (e.g., litter size) were recorded during the breeding season after AMH measurement to ensure a prospective analysis of the predictive value of AMH. Fertility in ewes, defined as the birth of singletons or twins, was recorded using the data obtained from the official central livestock registry at the University of Ljubljana, Biotechnical Faculty. All of the animals included gave birth to either one or two lambs, with the exception of a single ewe from the IP group, which did not produce any offspring.

### 2.2. Enzyme Linked Immunoassay for Anti-Müllerian Hormone

Circulating concentrations of AMH in the blood serum were determined using a commercial ovine AMH ELISA assay, per the manufacturer’s instructions (AnshLabs^®^, Webster, TX, USA). Two 96-well plates were used to analyze all of the samples. A serum sample from the lambs expected to have no circulating AMH was spiked with AMH to obtain concentrations of 2 ng/mL, 1 ng/mL, and 0.5 ng/mL. The spiked samples averaged 119 ± 8.4% of the expected concentrations. The intra-assay coefficient of variation (CV) averaged 3.9%, and the lowest point on the standard curve was 0.38 ng/mL, below which the samples were considered undetectable.

### 2.3. Statistical Analysis

The statistical analysis was conducted utilizing R 4.3.2 software [17], and *p* ≤ 0.05 was considered significant. The AMH concentration results were normally distributed after logarithmic transformation (*p* > 0.05). AMH concentrations between JS and IP were analyzed with a linear model in R package emmeans [18] to account for the different age groups in JS. To evaluate the influence of AMH concentration, age, and breed on pregnancy outcome, a generalized additive model was used as there was a non-linear relationship between the predictors and outcome. Data are presented as mean ± SE.

All of the animal experiments were approved by the Ethics Committee of the Veterinary Faculty of University of Ljubljana (No. 8-7/2021-2).

## 3. Results

Our study included 78 sheep of two indigenous Slovenian sheep breeds. In the JS group, 43 sheep were sampled. Their age ranged from 1 to 10 years, with an age of 5.37 ± 0.37 years. Serum AMH concentrations in JS ranged from 1.30 to 21.55 ng/mL, with a mean ± SE of 4.80 ± 0.63 ng/mL. In the IP group, 35 sheep were sampled. Their age ranged from 6 to 15 years, with an age of 10.32 ± 1.92. The serum AMH concentrations in IP ranged from 0.16 to 4.61 ng/mL, with a mean ± SE of 1.82 ± 0.19 ng/mL. The number of lambs born was 54 and the number of live births was 53, of which 11 pairs of twins were born. In JS, the average number of lambs per ewe was 1.23 ± 0.07 and in the IP group, the average number of lambs per ewe was 1.09 ± 0.05.

There was a significant difference in AMH concentration between the observed breeds (Figure 1A; *p* < 0.001). Due to the age range of the JS sheep included in the study, which could influence this result, they were categorized into three age groups. In the JS breed, the AMH concentrations exhibited a slight negative correlation with age, although this association was not statistically significant (R = −0.05, *p* > 0.05). No significant differences in AMH levels were observed across the three age subgroups (*p* = 0.752), with mean ± SE values of 4.98 ± 1.36 ng/mL (subgroup A), 5.28 ± 1.10 ng/mL (subgroup B), and 3.93 ± 0.70 ng/mL (subgroup C) (Figure 1B).

A weak negative correlation (R = −0.23, *p* > 0.05) was observed between age and AMH concentrations in animals ≥ 7 years across both breeds. However, within this age group, JS ewes exhibited significantly higher AMH levels (3.93 ± 0.70 ng/mL) than IP ewes (1.82 ± 0.19 ng/mL; *p* < 0.001).

There was a statistically significant difference (*p* < 0.001) in AMH concentration in the group of ewes with one or two lambs (Figure 2). The AMH concentration was higher in ewes lambing two lambs (7.65 ± 2.18 ng/mL) than in animals lambing one lamb (2.53 ± 0.16 ng/mL). The AMH concentration had a statistically significant (*p* < 0.001) effect on the number of born lambs, where JS breed and age group ≥ 7 years had a significantly higher number (*p* < 0.001) of lambs born (Figure 2).

## 4. Discussion

Anti-Müllerian hormone is an established marker for assessing ovarian reserve and shows a positive correlation with fertility in sheep, cattle, and other livestock species [1,4,19]. The present study provides valuable insights into the role of AMH in influencing fertility in two different sheep breeds in Slovenia, JS and IP, which are characterized by year-round and seasonal reproductive activity, respectively. Our results show clear, statistically significant differences in AMH concentrations between these two breeds.

In the JS breed, we found that animals with higher AMH concentrations were significantly more likely to give birth to twins. This finding aligns with past research. Increased AMH levels suggest greater follicular reserves and better reproductive capabilities, allowing multiple offspring per pregnancy [4]. Moreover, Turgut and Koca [20] showed that Romanov sheep with higher AMH concentrations (>320 pg/mL) had larger litters. A similar trend was seen in cattle. In both Bos indicus and Bos taurus calves, plasma AMH levels strongly correlate with the number of antral follicles, oocyte yield, and blastocyst production, regardless of gonadotropin stimulation [21]. In ruminants, higher levels of follicle-stimulating hormone (FSH), 17β-estradiol (E2), and AMH at the pre-superovulatory stage relate to increased embryo production. This matches observations in sheep, where more antral follicles and higher serum AMH levels indicate a greater ovarian reserve and fertility, where a higher number of antral follicles and higher serum AMH levels are considered indicative of a greater ovarian reserve and higher fertility potential [22]. This suggests that AMH could help select ewes with higher fertility in breeding programs, particularly in breeds like JS, where AMH levels strongly link to twin births. Conversely, the IP breed had significantly lower AMH concentrations overall, which could explain why twin births were less common in this group. Interestingly, these lower AMH levels were consistent with the literature suggesting that breed-specific genetic and physiological factors contribute to variations in AMH expression and consequently reproductive outcomes [23]. This supports the hypothesis that breed-specific genetic divergence drives AMH variability, with JS sheep likely possessing traits favoring higher ovarian reserve, while IP sheep genetically exhibit lower AMH and lower fertility [20]. Age is another factor that could have an influence on AMH levels in the IP breed. Most of the IP sheep in our study were older and it is well documented that age has a negative effect on AMH concentrations due to a natural decline in ovarian reserve [19].

This decline in AMH concentration as sheep age is consistent with observations in other species, where older females have a lower reproductive capacity, resulting in fewer offspring per pregnancy [4,23]. In cattle, it was found that calves have higher AMH levels than cyclic heifers, also suggesting an age-related decline in ovarian reserve—a trend that may also apply to sheep, although further validation is required. Their research emphasizes AMH’s value as a non-invasive fertility predictor, even in prepubertal animals, which supports our observations about AMH’s predictive power across different breeds [21].

Consequently, the lower incidence of twins in the older IP group could be attributed primarily to this age effect, as aging not only reduces the ovarian reserve, but also impairs the physiological environment required for multiple ovulations.

In contrast, in the JS breed, we had a more diverse age profile that allowed for comparisons between different age groups. Interestingly, age had no significant effect on AMH levels in this breed within the age range studied. Even though this may not necessarily apply to all age categories, it suggests that the JS breed has a more resilient reproductive system that maintains AMH levels and fertility potential as animals age, in comparison with IP breed, as there have been significant differences in AMH concentrations in both breeds ≥ 7 years old (*p* < 0.001). This resilience could be based on genetics and warrants further investigation to determine if selective breeding within this breed could increase longevity and productivity without compromising reproductive performance. However, it needs to be pointed out that the absence of younger IP sheep is a limitation of the present study, as it limits the ability to fully assess age-related variations in AMH levels and breeding potential; this aspect should be considered in future studies with a more balanced age distribution.

These breed-specific differences could be further confirmed by our methodology. The commercial AMH ELISA assay, originally developed for sheep and validated here using spike recovery experiments (119 ± 8.4% recovery), has shown sufficient accuracy for the detection of AMH in sheep blood serum. The stability of AMH levels across the estrus cycle, reported by Turgut and Koca [20], emphasizes the reliability of AMH measurements at a single time point in our study. While AMH concentrations in cattle vary with age [24], our assay demonstrated high precision (3.9% intra-assay variability) and a limit of detection (0.38 ng/mL) consistent with physiological AMH concentrations in small ruminants [25,26].

The methodological consistency of AMH quantification in different animal species strengthens its utility in fertility research. Xu et al. (2022) [27] emphasized the diagnostic utility of serum AMH as a predictor of multiple ovulation outcomes in sheep [27]. The pronounced AMH expression in the highly fertile JS breed, as found in our study, supports its utility as a selection biomarker for inclusion in elite breeding programs. Conversely, the lower AMH levels and suboptimal ovulation response in the IP breed could be targets for genetic or nutritional intervention, as suggested in studies of fertility modulation in sheep with different AMH phenotypes [20].

The results of our study highlight the importance of AMH as a predictive marker for fertility management in indigenous sheep breeds in Slovenia. Its use in breeding programs for JS sheep could represent a turning point, enable more precise selection of highly fertile ewes, and optimize overall flock productivity. However, in the IP breed, the lower AMH values suggest a different management approach. If the productivity of this breed is to be improved, attention should be focused on younger animals or on nutritional and hormonal interventions that could improve ovarian reserve and AMH levels.

Interestingly, some studies suggest that nutritional interventions, such as increasing energy intake prior to mating, can increase ovulation rates in sheep [28]. While these interventions may not fully offset the effects of age, they could serve as a strategy to improve fertility in the IP group and potentially mitigate the effects of lower ovarian reserve. However, it is worth noting that the AMH levels remained stable despite variations in diet or the presence of infections, suggesting that AMH serves primarily as a biomarker of ovarian reserve rather than a dynamic indicator of reproductive response to dietary changes or health status [19]. This stability suggests that AMH concentration is unaffected by such factors and, therefore, may not be effective as a marker of reproductive improvement when nutrition or other external variables are altered. On the other hand, a systematic review by Werner et al. (2024) highlights that environmental factors can significantly influence AMH levels in humans [29]. These findings are in line with previous studies in domestic ruminants [20,21], suggesting that metabolic health and nutrient distribution are fundamental to reproductive performance. In sheep models, metabolomic analyses have revealed several biochemical signatures associated with improved embryonic performance. In particular, phosphatidylcholines, which are critical for cell membrane structure and intracellular signaling, are consistently enriched in high embryo-yielding ewes in both induced and spontaneous estrus groups. These molecules may facilitate follicular development and oocyte competence, thereby promoting early embryogenesis [21]. Furthermore, metabolites linked to amino acid and fatty acid metabolism—such as pentadecanoic acid—are significantly associated with higher embryonic output, reinforcing the importance of systemic metabolic regulation in ovine fertility [27]. These insights underscore the importance of considering extrinsic influences when evaluating AMH as a biomarker in sheep. It is plausible that similar environmental modulation occurs in ruminants, potentially impacting both AMH levels and the associated metabolic signatures. Consequently, standardized experimental conditions and comprehensive phenotyping are essential for the reliable interpretation of AMH dynamics in livestock. Future studies should investigate the interactive effects of diet, environmental conditions, and genetic background on AMH production and metabolic phenotype, so as to refine its predictive value for fertility outcomes in ovine reproductive management. Beyond the immediate hormonal and age-related outcomes, our study opens a broader discussion of the interplay between genetics, environment, and reproductive outcomes. The genetic predisposition of JS sheep to higher ovarian reserves suggests a breed-specific evolutionary advantage that could be optimized by targeted breeding strategies. On the other hand, certain genetic disadvantages of the IP breed, such as lower AMH levels and a reduced incidence of twin births, raise the question of whether selective breeding or improved management strategies, such as enhanced nutrition, could effectively mitigate these issues. This interaction between genes and environment reflects the complexity of fertility management in agricultural species, where breed-specific approaches are required to maximize reproductive potential. Understanding these interactions and their effects on AMH expression could lead to more sophisticated breeding programs tailored to increase productivity while maintaining the genetic diversity and resilience of sheep populations.

## 5. Conclusions

Our study highlights the significant influence of the breed on AMH concentrations and reproductive outcomes in sheep. The higher AMH levels in the JS breed correlate with a higher number of twin births, suggesting a larger ovarian reserve—making them suitable candidates for breeding programs to maximize fertility. In contrast, the lower AMH values of the IP breed may reflect a reduced ovarian reserve or granulosa cell function, suggesting that targeted interventions are required, possibly through nutritional strategies or selective breeding management in order to enhance the overall reproductive capacity.

Importantly, AMH should be considered as a biomarker of ovarian reserve and not as a direct target for manipulation. Its utility in selection programs depends on interpreting levels within physiologically appropriate ranges.

Therefore, breed-specific reference ranges for AMH should be established to improve their use in fertility prediction. Future studies should aim to clarify the genetic determinants of AMH variability and evaluate how environmental and nutritional factors can support reproductive health in a way that indirectly contributes to more favorable AMH profiles.

## Figures and Tables

**Figure 1 animals-15-01332-f001:**
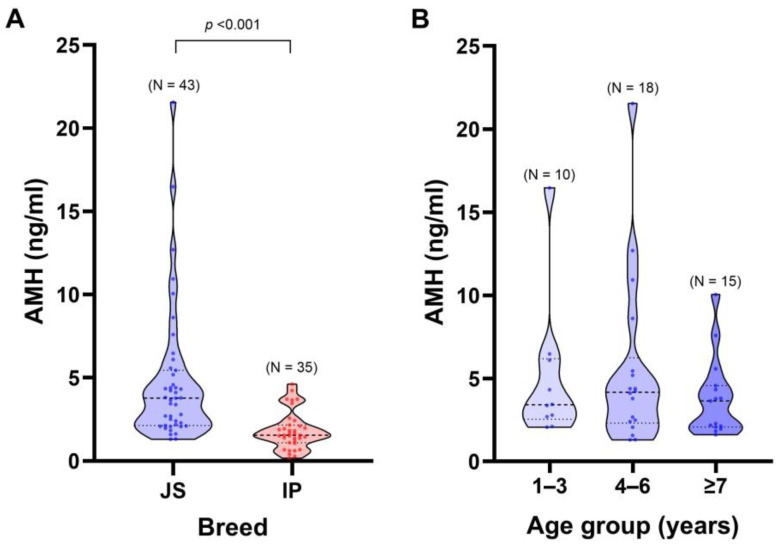
Determined anti-Müllerian hormone (AMH) concentration (ng/mL) in ewes. (**A**) Comparison of AMH concentration (ng/mL) between Jezersko–Solčava (JS) and Istrska pramenka (IP) breeds (*p* < 0.001). (**B**) Comparison of AMH concentration (ng/mL) between three age groups of the JS breed (*p* > 0.05) (corresponding with the text; A = 1–3 years, B = 4–6 years, C = ≥7 years).

**Figure 2 animals-15-01332-f002:**
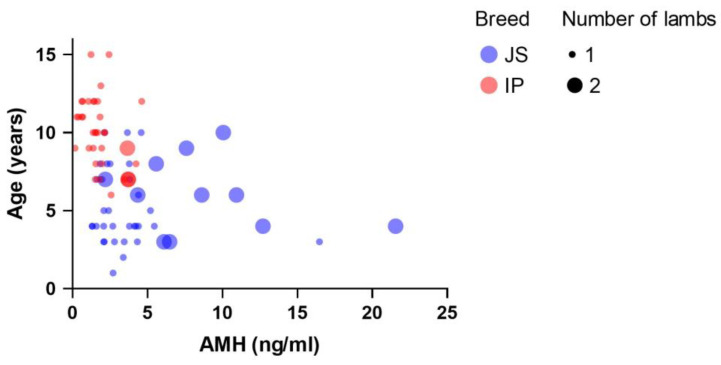
Bubble plot with the number of lambs and concentration (ng/mL) of AMH according to age; correlation coefficient between age and AMH in the entire sample was −0.47 and −0.31 in the subgroup ≥ 7 years.

## Data Availability

The original contributions presented in this study are included in this article, further inquiries can be directed to the corresponding author.

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
