# Peer review of "Comparative Analysis of Anti-Müllerian Hormone Concentration in Two Indigenous Slovenian Sheep Breeds"

_animals, 2025, doi:10.3390/ani15091332_

Round 1
Reviewer 1 Report
Comments and Suggestions for Authors
The manuscript „Comparative analysis of anti-Müllerian hormone concentration in two indigenous Slovenian sheep breeds” describes the AMH levels in non-seasonal Jezersko-Solčava and the seasonal breeder Istrska pramenka sheep, with higher AMH levels in JS and a positive correlation with the number of offspring.
The abstracts are clear and informative.
10: is expressed in females only in granulosa cells, mainly by growing follicles
19: produced only by granulosa cells, is a biomarker for ovarian
66/194: environmental factors not investigated in this study
69-72: The aim of this study was to evaluate AMH concentrations in IP and JS sheep and to investigate the possible relationships between AMH concentration, age and fertility outcomes (number of lambs).
Give more details in 2.1: Blood sample were collected all during fall (76) and then over the year (84)? How old were the IP? When after estrus detection were the samples collected? 84: estrus period=breeding season or estrus? How was the fertility measured? Mean number of lamps during live or total number? Last or further pregnancy? Make sure, that the ARRIVE guidelines are fulfilled.
89/90: “A serum sample from lambs expected to have no circulating AMH...”The AMH assays shows significant matrix effects. Therefore, this step is very welcome. But the blood of lamps might not suitable. As shown by Ali et al, 2017, Biol. Of. Reprod., Batista et al., 2016 Dom. Anim. Endo 56, or de Motais et al., 2024, Reprod. Dom. Anim. 59, have calves higher AMH levels than adolescents or adult cows, especially very young ones. Therefore, it is not clear, why the AMH levels of the older sheep are over the detection level or they might be underestimated. Please bring that point in the discussion.
115: describe the means and medians for Fig. 1 A/B in the text
Calculate correlation coefficient between age and AMH, maybe separately for the groups >7
118-120: Fig. 2 cannot be interpreted (repeat of Fig 1). It is not clear, how the lambs were counted. Were there ewes with three or no lamps (exclusion criteria?). What about the mean lambs/litter during their live? If the data are not normal distributed, medians should also be described.
183: group and potentially mitigate the effects of lower ovarian reserve.
192: The genetic predisposition of JS sheep to higher ovarian reserve
212: l “interventions that could increase AMH levels”. Do you really mean AMH levels? You suggest AMH as an indicator for ovarian reserve. In 183, 192, 212 you say that the AMH should be increased, that would mean the same ovarian reserve but higher secretion rates of AMH by the granulosa cells. AMH is an inhibitory substance and can have negative effect, if they are too high (Mossa, 2010, Reprod.; Mossa, 2019 J Anim Sci)
The discussion will benefit if more comparisons with studies about AMH in sheep would be cited. E.G.: Turgut et al., 2024, RDA have done a very similar study what can be directly compared. Xu et al., 2022, BMC Vet. Res., will give an answer for higher fertility, as well as Cushman et al., 2023, Anim Reprod. Sci.
Author Response
Thank you very much for taking the time to review this manuscript. Please find the detailed responses in the attachment file.

Reviewer 2 Report
Comments and Suggestions for Authors
Dear editors,
the present work was conducted in 2 slovenian indigenous breed of Sheep, and examines the influence of breed and age on the AMH concentration, also evaluating the possible relationship with fertilty outcomes (number of lambs). This approach derives from the need to find useful markers of fertility in these breeds, with the goal to lead to more sophisticated breeding programs tailored, increasing productivity while maintaining genetic diversity of sheep populations. The topic is attractive because it provides more information about the endocrinology of sheep reproducton. Although the references are specific, in my opinion, they are insufficient for discussing the topic in a focused manner. Thus, the discussion could be more deeply developed. I suggest adding some pertinent references.
Summary and Abstract: They are well written and summarize the information contained in the main text without repetitions. They provide background by placing the addressed question in a broader context, and include all the elements that should be present in an abstract and summary aligned with the editor’s requirements
Title, Keywords, and Introduction: The title is attractive and adequate for the content of the paper. However, the acronym rather than the full name should be used when referring to genes or proteins. In the keywords section, I suggest you to add some relevant words like: "aging", "ovarian reserve" "prolificacy", “fertility”. The introduction is relevant and effectively contextualizes the main aspects of the topic. The aim is clearly stated, but the hypothesis should be included. Lines 10 and 19: the acronym 'AMH' should be explained only the first time it appears. The same applies to the acronyms 'IP' and 'JS' in this section and in the main text. Lines 67–68: the aim is specified in the following sentences; please revise this redundant sentence.
Materials and Methods: The study design is very simple but well developed. Retitle the paragraph 'Ewe' as 'Animals and Sampling'. Some details are missing in this section. I suggest adding more information about the sheep involved in the study (Were they treated with any medication during the study? Were they suffering from any pathology? It should be specified that the ewes were mated and became pregnant. How many of them had one or two lambs? This information is important to correlate these parameters with AMH levels. Also, specify the type of farming system in which they were raised). Line 80: please add the letters “A, B, C” to the age groups (by the side of the ranges) and use them also in the figures. Lines 83-84: how many sample for each animals? At which time of the day (same hours)? For the JS breed you sampled during the short season of reproductive activity, but for the IP during whole year: it would be interesting to add information about the month of the year in which the study was conducted, as well as the average environmental temperature and humidity of these period. All this information is important in a physiological and endocrinological study. P value should be written in italic and ≤, fix it in the text. Lines 102-103 and Lines 106-107: please, move these sentences to the animal’s paragraph. Line 105: no need to explain acronyms again.
Results: They are logically presented and accompanied by clear figures. Figure 2: The plots are the same as those in Figure 1. Please add the correct ones.
Discussion and Conclusion: The discussion, interesting and overall well done, follows a logical line and presents persuasive interpretations, but the current knowledge on the topic should be more properly summarized, by adding further references to make your interpretations more solid. In fact, the results should be discussed in more depth, comparing them with other bibliographic sources and with other specie,s such as cows, as the discussion is currently too brief and supported by too few references. Additionally, some concepts need to be clarified or revised. I have some considerations to make. Line 157: To support this statement, you should include a comparison with younger ewes, evaluate their AMH levels, and assess whether they are significantly different from older ones. Line 160: the same consideration applies. Not knowing whether there is a decline, you should compare the values with those of younger ewes of the same breed. this can only be hypothesized. Therefore, this represents a limitation of your study and should be acknowledged. After all, as you state in line 167, in the JS breed there is no age-related decline. The same might occur in the IP breed. This might explain why the IP breed is less prolific. Please revise the previously mentioned sentences, to avoid misleading conclusions. In addition, you specified that you sampled during the short season of reproductive activity for the JS breed, but during whole year for the IP. Did you consider the impact of the different environmental conditions (seasonal effect) and food resources (food quality and intake) during the short period of the reproductive season compared to the whole year on AMH concentrations? It is another limitation of your study and should be discussed in this section, especially in light of the fact that your results aim to provide tools for improving fertility through changes in management and nutrition.
Overall, the paper is well written and, since it adds useful information about endocrinology of sheep reproduction, it deserves to be published after minor revisions.
Author Response

(The authors gave the same response as above.)

Reviewer 3 Report
Comments and Suggestions for Authors
Minor corrections required which have been marked in manuscript. However, following may be answered.
In line 116-117, the comparison of AMH concentration in two breeds in > 7 years age group has been presented. However, it has not been discussed in discussion portion. The same may be justified and presented.
Figure 2 is required to be changed. The legend and text says that Fig. represents Bubble plot between number of lambs v/s anti-Müllerian hormone (AMH) concentration. But the figure itself is repetition of Figure 1.
Authors have not included animals of IP breed from age group of 1-3 years and 4-7 years which is more pertinent as the authors attempt to compare breeding capabilities vis-a-vis AMH concentration. Non inclusion of these groups need to be justified.

Author Response

(The authors gave the same response as above.)

Round 2
Reviewer 1 Report
Comments and Suggestions for Authors
Thank you for your conscientious implementation of the recommendations. The correlations between AMH and age should be described in one sentence in the results section. It is still not clear to me whether the fertility data come from the last or the following season, i.e. whether AMH can be considered retrospectively or prospectively. Otherwise the article can be published.
Author Response
Thank you very much for taking the time to review this manuscript. Please find the detailed responses below and the corresponding revisions/corrections highlighted/in track changes in the re-submitted file.
Comments 1: Thank you for your conscientious implementation of the recommendations. The correlations between AMH and age should be described in one sentence in the results section. It is still not clear to me whether the fertility data come from the last or the following season, i.e. whether AMH can be considered retrospectively or prospectively. Otherwise the article can be published.
Response 1: The following sentences were added into the manuscript in order to provide clarification:
- Lines 139-150:
" In the JS breed, AMH concentrations exhibited a slight negative correlation with age, though this association was not statistically significant (R= -0.05, p > 0.05). No significant differences in AMH levels were observed across the three age subgroups (p = 0.752), with mean ± SE values of 4.98 ± 1.36 ng/mL (subgroup A), 5.28 ± 1.10 ng/mL (subgroup B), and 3.93 ± 0.70 ng/mL (subgroup C) (Figure 1B).
A weak negative correlation (R= - 0.23, p > 0.05) was observed between age and AMH concentrations in animals ≥ 7 years across both breeds. However, within this age group, JS ewes exhibited significantly higher AMH levels (3.93 ± 0.70 ng/mL) than IP ewes (1.82 ± 0.19 ng/mL; p < 0.001)."
- In the Methods section, we now explicitly state (Lines 101-103):
"Fertility results (e.g. litter size) were recorded during the breeding season after AMH measurement to ensure a prospective analysis of the predictive value of AMH."
Reviewer 3 Report
Comments and Suggestions for Authors
The authors have replied to the queries raised
Author Response
Comments: The authors have replied to the queries raised.
Respond: We thank the reviewer for their comments and are pleased to hear that the queries raised have been addressed to their satisfaction. We remain available for any further clarifications if needed.